# Disentangling Environmental, Economic, and Technological Factors Driving Scallop (*Argopecten purpuratus*) Aquaculture in Chile

José Bakit [1,2,3,*] , Gonzalo Álvarez [1,4,*] , Patricio A. Díaz [5] , Eduardo Uribe [1], Rodrigo Sfeir [6] , Sebastian Villasante [2,3], Tomas Gabriel Bas [6], Germán Lira [7,8], Hernán Pérez [9], Andrés Hurtado [10] , Raúl González-Ávalos [11] and Jose Castillo-Venenciano [12]

1 Departamento de Acuicultura, Facultad de Ciencias del Mar, Universidad Católica del Norte, Coquimbo 1780000, Chile
2 EqualSea Lab-Cross-Research in Environmental Technologies (CRETUS), Departamento de Economía Aplicada, Universidad de Santiago de Compostela, 15705 Santiago de Compostela, Spain
3 Campus Do Mar, International Campus of Excellence, 15705 Santiago de Compostela, Spain
4 Centro de Investigación y Desarrollo Tecnológico en Algas (CIDTA), Facultad de Ciencias del Mar, Universidad Católica del Norte, Coquimbo 1780000, Chile
5 Centro i~mar and CeBiB, Universidad de Los Lagos, Casilla 557, Puerto Montt 5480000, Chile
6 Escuela de Ciencias Empresariales, Universidad Católica del Norte, Coquimbo 1780000, Chile
7 Laboratorio Central de Cultivos Marinos, Facultad de Ciencias del Mar, Universidad Católica del Norte, Coquimbo 1780000, Chile
8 Programa Magister en Acuicultura, Facultad de Ciencias del Mar, Universidad Católica del Norte, Coquimbo 1780000, Chile
9 Centro de Innovación Acuícola AquaPacífico, Larrondo 1281, Coquimbo 1780000, Chile
10 Instituto de Políticas Públicas, Universidad Católica del Norte, Coquimbo 1780000, Chile
11 Maritime Engineering Laboratory, Universitat Politècnica de Catalunya-BarcelonaTech, 08034 Barcelona, Spain
12 Escuela de Ingeniería, Universidad Católica del Norte, Coquimbo 1780000, Chile
* Correspondence: jbakit@ucn.cl (J.B.); gmalvarez@ucn.cl (G.Á.)

**Abstract:** The boom-and-bust trajectory of the *Argopecten purpuratus* industry in Chile shows the progression from resource extraction (fishing) to production (aquaculture). This paper analyses the effects of environmental, economic, and scientific–technological factors. The influence of each factor on scallop production in Chile was reviewed for the period between the 1980s and 2020. The evaluation of the effects allows the visualisation of the industry's productive evolution and reveals the current challenges. The occurrence of abrupt environmental disturbances, commercialisation under imperfect market configurations, and public and private efforts in scientific and technological advances have acted favourably on scallop production. However, an industry mainly focused on prices and high production volumes did not devote much effort to develop low-cost climate-resilient technologies. Today, economic challenges must be addressed by helping to reduce production costs and add economic value to products and by-products. Our results show that the industry must focus on low-cost technologies, the use of renewable energy, and the circularity of its processes. The environment ensures the capture of natural seeds and their adaptation to climate change. These challenges must not lose sight of the emerging effects of the COVID-19 pandemic.

**Keywords:** *Argopecten purpuratus*; scallop; boom and bust; aquaculture; production; Chile; market structure; challenges

## 1. Introduction

The extraction of *Argopecten purpuratus* in Chile for commercial purposes increased sharply in the 1980s [1–5] as a consequence of an intense El Niño (ENSO) event [6–8]. This period registered increases in scallop landings that triggered the overexploitation of



the resource, forcing the Chilean Undersecretariat of Fisheries (*Subsecretaría de Pesca*) to prohibit its extraction from natural beds and authorising only the collection of wild spats for aquaculture [9]. The establishment of a regulated management system that imposed territorial conditions on production [10] and the enactment of the General Law on Fisheries and Aquaculture (*Ley General de Pesca y Acuicultura*, LGPA) [11] are key milestones that allow the consolidation of scallop aquaculture [1,12,13]. In the mid-1990s, Chile had a production base of 27 companies—*A. purpuratus* culture centres, nine hatcheries, and ca. 3000 culture lines at sea in the Atacama and Coquimbo regions, which accounted for 98% of the Chilean production [14], thus becoming the world's second-largest producer of scallops [5]. Landings of *A. purpuratus* exhibited an increasing trend until the 2000s, when production stagnated, followed by decreased production activity.

Transitions towards the expansion and collapse of different fisheries have been studied, [15–18] particularly in the genus *Pecten*, such as those related to overfishing [16,19,20], large variations in recruitment conditions [21,22], exogenous effects on species livelihoods [23], seasonal climatic conditions [24], and gradual climate change, or a combination of the above [25].

In Chile, the expansion of scallop production, first as fishery and then as aquaculture [3,4,26], is attributed to environmental, technological, commercial, and political–economic factors. The environmental conditions that favoured scallop production were mainly natural beds, natural recruitment, water temperature, and the carrying capacity of its bays [22,27–30]. Technologically, the transfer and adaptation of the Japanese suspended available culture system appears as the driving force of scallop aquaculture [3,21]. Finally, in commercial terms, demand in foreign markets, high scallop prices [1,2] and national commodity export policies [31–33] are pointed out as catalysts for production activity.

However, under specific scenarios, it was possible to visualise that these four aspects also allowed the decrease of scallop production in Chile. Environmentally, the high variations in seed recruitment [21–34], the effects of the different depths of each bay [35], and the need for sanitary water certification [32] stand out. Technologically, it has focused mainly on restocking, genetics, and production systems [36], leaving aside the development of efficient technologies and systems for economic decision making. Economically, Chilean scallop producers have not been protagonists in commercial activity because decisions on quality and size are concentrated on the buyer and not in their best interests [37], adding to the large percentage of production destined for a single consumer market, configuring an imperfect market. Finally, surveillance and control policies have not fulfilled their functions in the face of illegal harvesting [26,34].

In addition, several authors pointed out the irruption of *A. purpuratus* production from Peru as the cause of the fall in the international price of the product and, indirectly, the collapse of the industry's production and exports in Chile. The reasons given were summarised as high production/landing volumes, low production costs [1,3,5,37], and the higher growth rate of scallops [38], which allows scallop harvest sizes to be reached in approximately eight months [39]. The combination of the mentioned effects provided Peruvian producers more competitive conditions vis-à-vis their Chilean counterparts and returns on investment in less time. This last point reduces the risk of the aquaculture activity to deal with abrupt environmental disturbances, enabling investors to demand a lower return on the business before investing or upgrading [12,40,41]. These situations create an unattractive scenario for investors in Chile, who seek to minimise the risk and maximise the benefits of the business, consequently leading to a lower production capacity.

To contribute to the knowledge of the Pacific scallop, this study aimed to analyse in depth the effects of economic, environmental, and technological factors on the productive performance of the industry in Chile. It also defines the present and future challenges to produce *A. purpuratus* in Chile. Thus, the aspects discussed in this study integrate research conducted on economic and societal aspects [1], production arts [5,42], household economics (i.e., the microeconomics of production) [12], and resource management [26,43].

## 2. Materials and Methods

### 2.1. Data Collection

An exhaustive literature review was carried out to identify the environmental, productive, commercial, and political–economic antecedents of *A. purpuratus* production in Chile. Sources ranging from scientific literature and patents to state reports or historical accounts allowed us to establish the date of occurrence and duration of environmental or economic events that may have affected production.

Science-Technology: Published scientific literature was analysed by reviewing international journals available in the Web of Science and Scopus databases. The selection was made from scientific articles, book chapters, and oral presentations in published conference proceedings [44], that mentioned "*Argopecten purpuratus*" in their title, keywords, or abstracts. The selection covered papers published in English, Spanish, and Portuguese between 1980 and July 2022. For the collected literature, information on five variables was recorded in previously defined categories that allowed comparison (Table 1). In addition, to deepen the research focus of the papers, dimensions and subdimensions were defined to cover the different production phases of *A. purpuratus* (Table 2).

**Table 1.** Variables and categories for data collection from the scientific literature review.

| Data Variable | Description |
|---|---|
| Type of document | Article \| Book chapter \| Conference paper \| Proceeding paper \| Communication |
| Type of paper | Conceptual \| Conceptual review \| Review \| Empiric \| Data paper |
| Type of data | Survey \| Survey–Official data \| Official data \| Experimental \| Modelling \| N.D. * |
| Type of analysis | Qualitative \| Qualitative–Quantitative \| Quantitative |
| Scale | Global \| National \| Regional \| Local |

\* Data not available.

**Table 2.** Dimensions for data collection from the research/patenting literature review.

| Dimensions | Sub-Dimensions |
|---|---|
| Reproduction | Gonadal \| Spawning \| Condition \| Others |
| Larval culture | Larvae \| Survival \| Growth \| Others |
| Environment | Recruitment \| Foods \| Toxins \| Pathogens \| Pollutants \| Carrying capacity \| Others |
| Technology culture | Culture systems \| Biomass production \| Hatchery \| Recirculating aquaculture systems (RAS) \| Others |
| Products | Nutritional \| Subproducts \| Others |
| Governance | Statistical \| Management \| Others |
| Commerce | Markets \| Supply chain \| Costs production \| Exports \| Others |

Data on implemented projects were also collected from governmental institutions that support Research and Development (R&D). The funding institutions reviewed were the Fund for the Promotion of Scientific and Technological Development (e.g., FONDEF, FONDEF-IDEA, and FONDECYT), CORFO Innovation Fund (INNOVA), the Fisheries and Aquaculture Research Fund (e.g., FIP and FIPA), and the Innovation Fund for Regional Competitiveness (FIC-R), and were classified according to the project's objective: economic, technological, or environmental projects. Since the projects had a broader scope than a scientific article, categories were generated to allow the initially defined classes to be combined. Data from productive development projects promoted by other governmental institutions (e.g., SERCOTEC) were not considered because their focus is to improve business management rather than R&D. Data on projects carried out by companies with their funds are unavailable.

Active patents assigned were identified by applicant and country using four keywords related to the species: *Argopecten purpuratus*, Chilean Scallop, Peruvian Scallop, and Purple Scallop. Thirteen web-based public patent databases were searched with global records: EspaceNet, Lens, Google Patent, FPO, and WIPO; and by country: LatiPat, European Patent, Invenes Interpatent, USPTO, Kippris, CNIPA, INDECOPI, and INAPI. A total of

420 patents were found; however, by using universal publication codes corresponding to each patent, it was possible to eliminate both repeated and inactive patents. Patent targets were classified according to the dimensions of the different production stages of *A. purpuratus* (Table 2).

Environment and landings: Scallop landing data (in metric tons) from Chile and Peru were collected from the Food and Agriculture Organization of the United Nations online statistical series [45], indicating the species "Peruvian calico Pecten", for the 1982 to 2021 period. The Oceanic Niño Index (ONI) was obtained from the Climate Prediction Centre [46]. Historical data from Niño/Niña episodes in the South Pacific were collected and analysed.

Economics: Trade and market data for *A. purpuratus* from Chile were obtained from Prochile and Aduanas (Customs) [47,48], specifying the quantities and destination markets in which scallops were traded, as well as the prices (USD) achieved. International trade prices between 1992 and 1998 were collected by Sfeir et al. [49]; trade data from Peru were gathered from Promperu [50], and the series of fishery and aquaculture statistical yearbooks from the Peruvian Ministry of Production [51], where a search for *Vieras* and *Concha de Abanico* (as *A. purpuratus* is known in Peru) yielded statistics, prices, and destination markets. The export values and prices quoted in this study were not adjusted for inflation.

### 2.2. Analysis of Economic Data

The economic analysis in this section was conducted by identifying four aspects considered relevant in economic or commercial terms: (i) the destination markets for scallop production, (ii) the effect generated by Peruvian production, (iii) the marketing prices of exported products, and (iv) the episodes of relevance and impact on the national and international economy. The data were processed and graphed using Excel® spreadsheets.

To determine the significant difference in the prices obtained by Chile and Peru for their exports, a t-test was used ($p = 0.05$). The analysis was conducted for the period between 1998 and 2020, when both countries showed consolidated production. Regression analysis was used to establish the relationship between scallop productions in both countries. For this purpose, the relationship $R_i$ (Equation (1)) was determined and evaluated for 1998–2020, as follows:

$$R_i = \frac{P_i^{Ch}}{P_i^{P}} \tag{1}$$

where $P_i^{Ch}$ expresses Chilean production in the year $i$ and $P_i^{P}$ expresses Peruvian production in the same year.

The temporal evolution of prices in Chile and Peru between 1993 and 2020 was tested using a time series analysis. The Augmented Dickey Fuller (ADF) test is a common statistical test used to determine whether a given time series is stationary [52]. The null hypothesis of the ADF test assumes the presence of unit root, that is, $\alpha = 1$, and the $p$-value obtained should be less than the significance level (0.05 or 0.01). In addition, a Johansen cointegration test [53,54] was applied because of the non-stationary nature of Peru's price time series. The Johansen cointegration test allows us to show whether two or more time series move in the same trend over time, with stable differences between them. Thus, this set of non-stationary time series is cointegrated if there is a linear combination of these series that is stationary. All statistical procedures were carried out using the statistical and programming software R 2.1.12 [55], package "tseries" [56] available through the CRAN repository www.r-project.org/ (accessed on 20 October 2022).

### 3. Analysis and Discussion

#### 3.1. Scientific and Technological Development Factors and Scallop Production

3.1.1. Scientific Publications

In total, 309 scientific publications were reviewed and categorised for the 1980–2022 period (Figure 1). By type of document, scientific articles (89.6% of the total publications) largely predominated, followed by conference proceedings (6.8%), with a low contribu-

tion from the rest of the categories (e.g., book chapters and communication). Empirical approaches (93.2%) outnumbered conceptual and conceptual reviews (3.9% and 1.6%, respectively); experimental data (81.9%) were the most used, followed by data derived from modelling (5.5%) and official data (4.2%), while quantitative analyses (97.1%) far outnumbered qualitative analyses or those combining both. The national scope (62.8%) of the studies addressed research conducted mainly in Chile, Peru, and China; locally (25.9%), the studies addressed research conducted in selected locations (e.g., Tongoy, La Rinconada in Chile and Sechura, Independencia in Peru), and those with a global scope represented 5.8% of the total contributions.

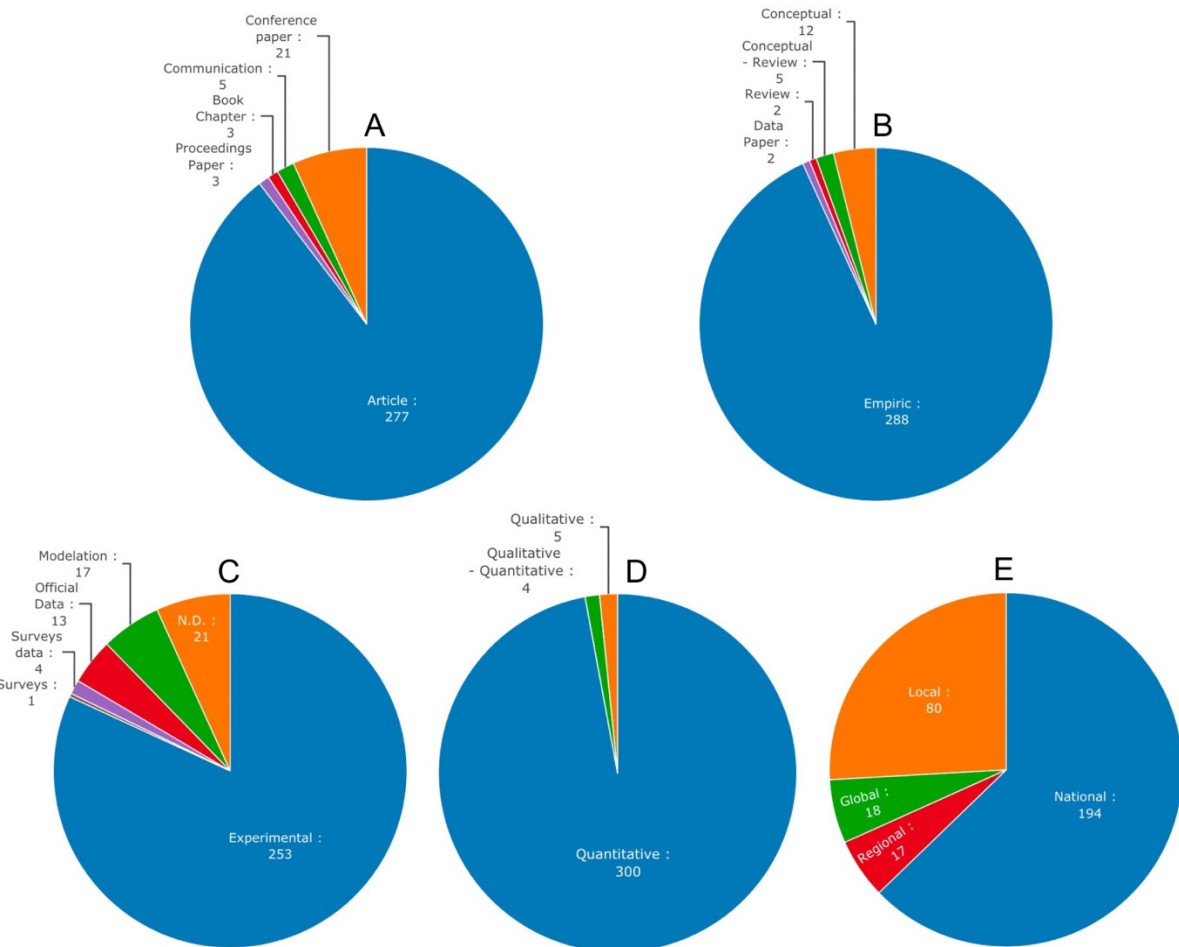

**Figure 1.** Analysed scientific publications (n = 309) from 1980 to 2022 (July) on *Argopecten purpuratus* by (**A**) type of document, (**B**) type of paper, (**C**) type of data, (**D**) type of analysis, and (**E**) geographical scale.

By thematic areas, our review's results on *A. purpuratus* (Figure 2A) show that the key topics addressed were the environmental issues of scallop production (41.4%), followed by reproduction (18.1%), culture technologies (11.0%), products (10.7%), larval culture (9.4%), governance (5.8%), and trade (0.6%). When analysing the temporality of the publications (Figure 2C), a boom-and-bust trend in the aquaculture industry was found, starting with the shift of the studies from commercial fisheries to aquaculture, the consolidation of the industry, and its subsequent decline [57]. For instance, Bustos-Gallardo and Prieto [31] indicate that, in the boom-and-bust dynamics, analyses were privileged depending on their stage of production. First, studies focused on the resource productivity to improve economic revenues. Second, they gave way to studies on mass production, management, and the causes and effects of this production on the environment. Finally, during the crisis,

more attention was paid on the study of how to contain the effects that caused the decline and recovery strategies.

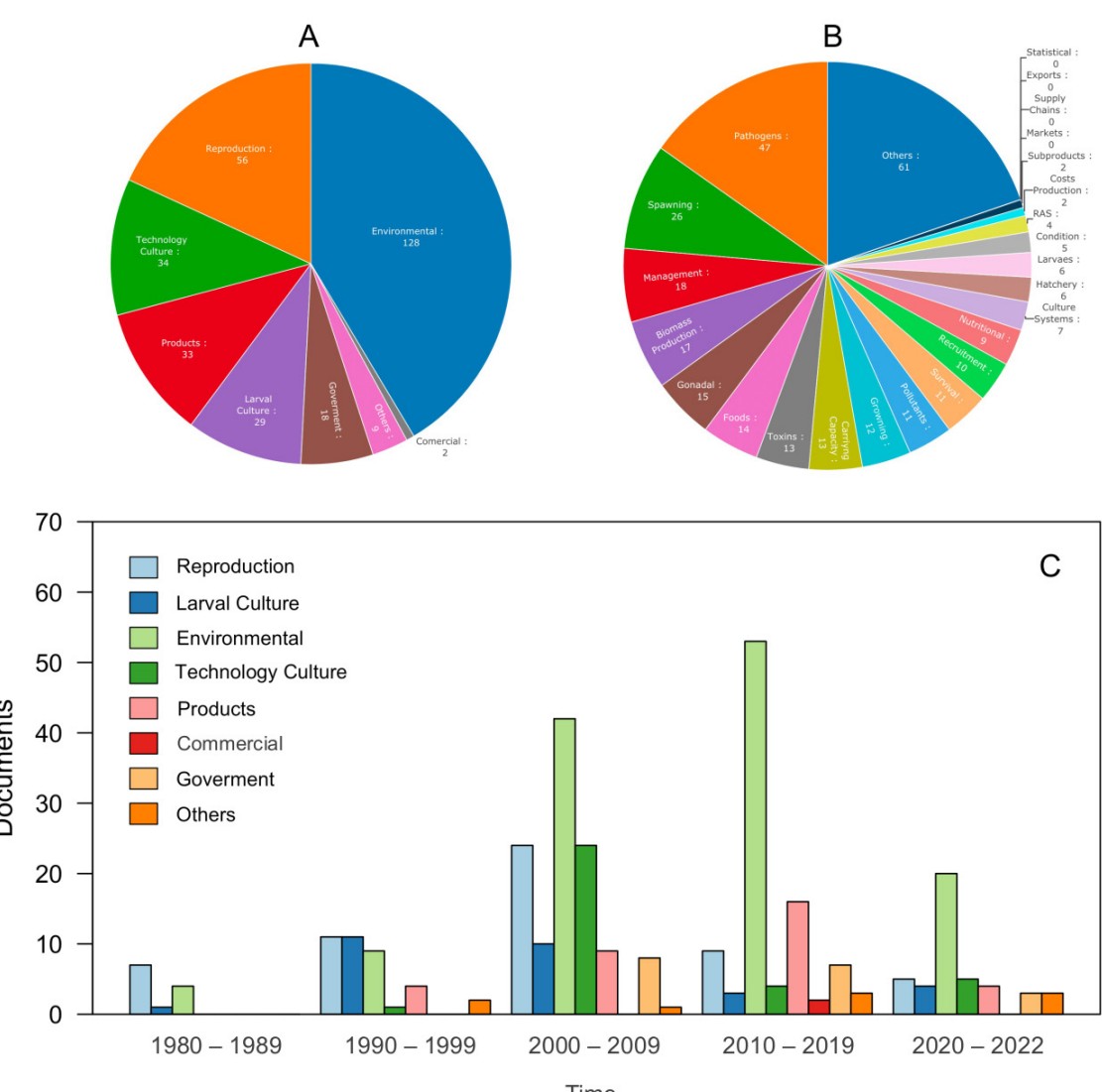

**Figure 2.** Analysed scientific publications (n = 309) from 1980 to 2022 (July) on *Argopecten purpuratus* by (**A**) research area, (**B**) research sub-area, and (**C**) temporal distribution of studies.

By decades, our results reveal that in the 1980s, the studies published on *A. purpuratus* (12) were related to reproductive (e.g., gonadal, spawning, and conditioning) and environmental (e.g., recruitment and food) issues. During the 1990s, the articles (38) continued referring to the previously studied themes and added larval culture and final product. By the 2000s, with a consolidated *A. purpuratus* industry in Chile and Peru, the articles (118) mainly addressed environmental issues (e.g., pathogens, toxins, carrying capacity, and upwelling), culture technologies (e.g., biomass production, hatchery, and RAS), reproduction, products, and incipient studies on governance systems required for the management of both marine resources and ecosystem services. In the 2010s, environmental, product, and governance issues were addressed in the published articles (97). During the 2020s, 44 articles have been published thus far, including studies on the effect of climate change and those looking at the circularity of scallop aquaculture through the search for by-products. However, our review also shows that commercial issues (e.g., markets, production costs, and value chain) related to the *A. purpuratus* production are still largely scarce (Figure 2B).

### 3.1.2. Research and Development Projects

Our results show that there is a predominance of technological and species farming issues (37.8%), environmental issues (11.1%), a mixture of them (15.6%), and only a few projects focused on economic aspects (2.2%) (Figure 3). The number of implemented projects that address all three areas in an integrated approach is 26.7%, mainly due to the incorporation of management and regulatory issues.

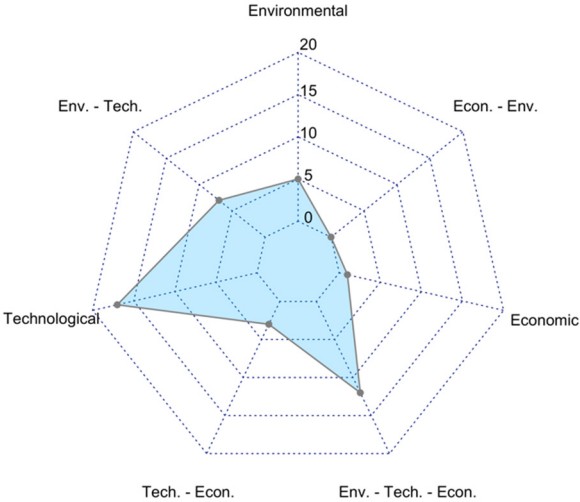

**Figure 3.** Funded *Argopecten purpuratus* projects, period 1990–2020.

The state mainly carries out R&D investment (e.g., FONDEF, FONDECYT, INNOVA, and FIPA), and private actions to finance R&D do not reach 25% [36,49]. Globally, there is a tradition of private innovation development based on state support [58]. For instance, of the total number of projects, FIPA was awarded 28.9%, INNOVA 22.2%, FONDECYT 20.0%, and FIC-R 15.6%. The innovation development based on state support is also seen in the agriculture sectors, where some authors such as Huffman and Just [59] were not optimistic about private companies funding research in R&D centres if the intellectual property protection mechanism is not strengthened.

### 3.1.3. Patents

The 159 patents we identified in our review were distributed across 10 countries, with an additional one belonging to the European Union. Figure 4B shows that the leading patenting country is China with 96 patents (60.4%), followed by the United States (33; 20.8%), Peru (11; 6.9%), and Chile and Japan (5 each; 3.1%). These results show the low interest of companies and R&D centres in Chile in patenting technological innovations for scallop aquaculture.

When analysing the number of patents per country and year (Figure 4C), we found that countries started patenting sporadically between 2004 and 2006. China consolidated its position as the country with the highest number of patents granted, growing from 12, 17, and 19 patents in 2012, 2014, and 2016, respectively. The United States started patenting in 2008 and peaked at 11 patents in 2017. A reduced and sporadic number of patents were observed for the rest of the countries with patents granted. The fact that all the patents found have a grant date within the last two decades is favourable for patent holders because innovations depreciate and lose value over time [60].

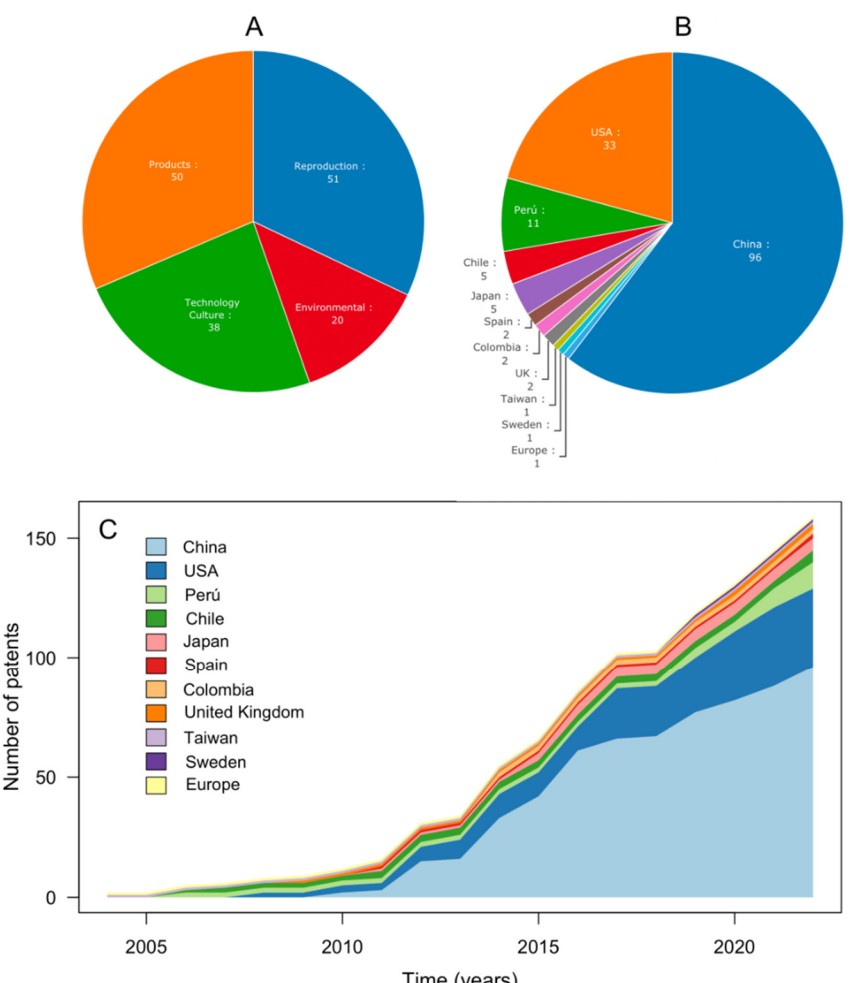

**Figure 4.** Analysed granted patents (n = 159) from 1998 to 2022 (July) on *Argopecten purpuratus* by (**A**) number of patents by area, (**B**) number of patents by country, and (**C**) temporal evolution of patents.

Figure 4A shows that patents were intended to address issues related to breeding (32.1%); products (31.4%), where patents on the final product and by-products predominate; cultivation technology (23.9%); and finally, environmental issues (12.6%) mainly for the use, control, and prevention of pathogens, toxins, and pollutants. No patent registrations in the other dimensions were found. In Chile, patents are registered for reproduction, culture management, and by-products.

### 3.2. Relationship between Environmental Factors and Scallop Production

#### 3.2.1. El Niño/La Niña—Southern Oscillation (EN/LN)

Intense El Niño/La Niña—Southern Oscillation (EN/LN) episodes have been widely studied and indicated as one of the main causes that affect *A. purpuratus* production. Specifically, this refers to water temperature, gonad indices, larval abundance, and juvenile recruitment [6–8,61–63]. Warm phase (cold phase) EN (LN) records were observed between October and February, except for the EN episode recorded in 1982 between August and September, which resulted in a ten-fold increase in historical landings in 1984. The relationship between EN/LN and *A. purpuratus* production in Chile shows an increase in landings following EN events (e.g., 1982, 1992, and 2002) and decreases in landings following LN events (e.g., 2007 and 2010) (Figure 5). The effects of EN/LN on production were observed 18 months after the event, which coincided with the duration of the productive cycle [37]. Exceptions were observed in 1987 and 1988.

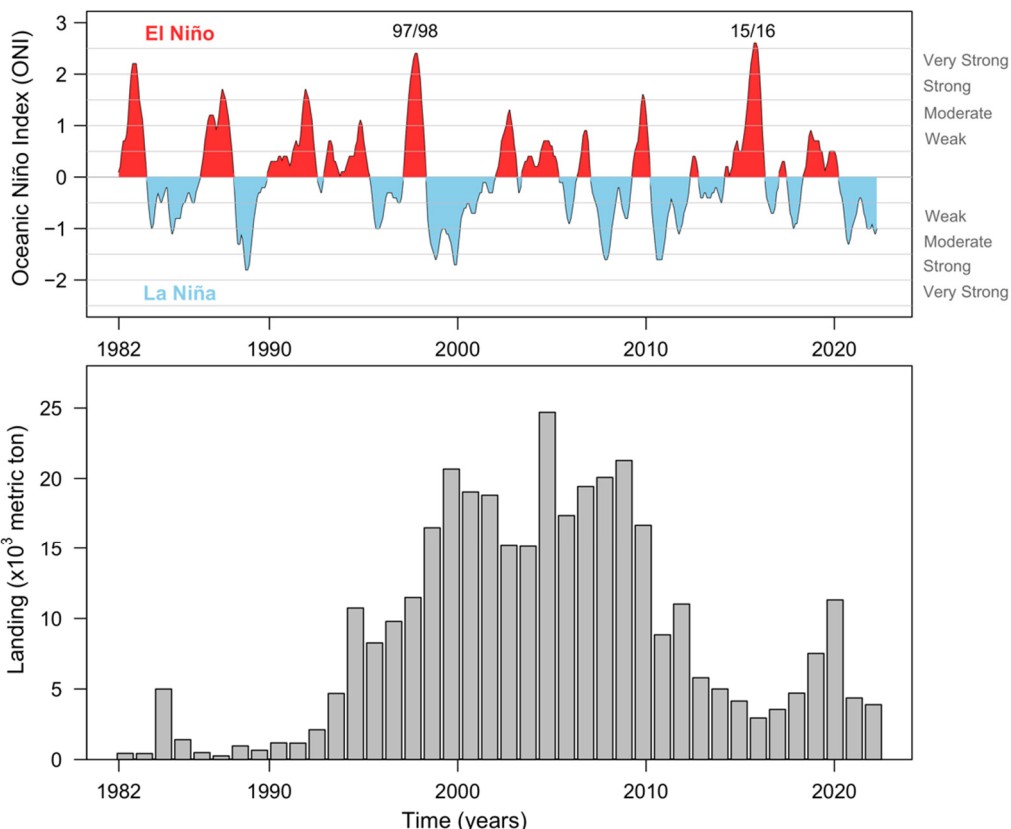

**Figure 5.** Production of *Argopecten purpuratus* in Chile vs. Oceanic Niño Index (ONI).

Subtropical areas (e.g., northern Chile) positively affect food availability in the absence of EN/LN, favouring fouling formation [64]. *Polydora* sp. and *Ciona intestinalis* cause considerable economic damage to scallop producers [64,65]. The negative effects are manifested in two dimensions: (1) the deterioration and loss of culture systems, which led to the collapse of almost 20% of the culture units in Bahía Inglesa due to attached *C. intestinalis* [66], and (2) the delayed growth of *A. purpuratus* in these fouled systems [65]. Official records indicate that the largest episode of Polydora infestation in *A. purpuratus* occurred in 1990 in Bahía Tongoy, resulting in high mortality of cultured (45%) and seed (80%) specimens, and those of commercial size had 20% weight loss and 30% increase in labour costs for processing [65].

The effects of EN (LN) events on scallop production were evaluated in the context of fishing and aquaculture stages of *A. purpuratus* production. Figure 6 shows the ONI, the years of occurrence, and its effect on landings after approximately 18 months. The effects of EN/LN are clearly observed in the fishery stage because the response of landings was closely related to fishing effort and stocks in natural beds [8,26,67]. The decrease in landings recorded in 1986 is indicated by the declination of the fishery stage [13–67] and forced the authorities to declare a temporary and then indefinite ban on the harvest of *A. purpuratus* [68]. This situation was studied in the past, and divergent conclusions were reached on the low population renewal of the resource, with some of them claiming it as a cause of overexploitation [26,63]. Yet, other studies pointed to increases and changes in the food preferences of predators [69] or the natural variability of the population [70]. Another study taken into account in this discussion, which included elements of previous studies [67], is related to the LN event seen in late 1984.

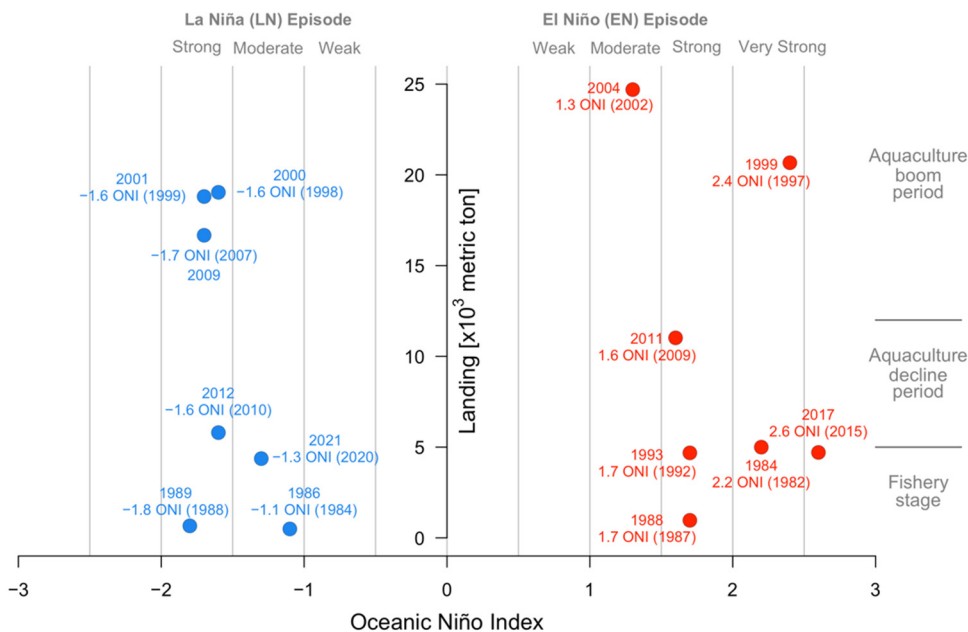

**Figure 6.** Effects of intense EN/LN events on the production of *Argopecten purpuratus*.

In contrast, in the aquaculture phase (i.e., from the 1990s onwards), landings reached 24,697 metric tons in 2004, almost five times the maximum landings recorded in fisheries. In aquaculture production, some population variables can be controlled, moving from the uncertainty of complex natural systems to the risk of event occurrence [13]. Thus, EN/LN events conditioned aquaculture farmers' production and control capacities. Figure 6 shows how EN events amplify the response of landings during the boom period (aquaculture), mainly because of the availability of culture systems, boats, and post-harvest facilities. For LN events, the response was attenuated, mainly because of low natural seed recruitment despite seed production from hatcheries.

### 3.2.2. Harmful Algal Blooms

Harmful algal blooms (HABs) have been detected at different aquaculture shellfish sites along the coast of Chile [5,71,72], forcing the establishment of sanitary protocols for domestic and export consumption [73]. To comply with European Union regulations and Food and Drug Administration (FDA) requirements, Chilean authorities established the Bivalve Mollusc Sanitation Program (PSMB). The results of this program were positive for scallop production because of the non-occurrence of poisoning cases while also avoiding adverse trade effects (e.g., harvest restriction, reduced employment, decreased investments, and changes in consumption habits) [73]. During the 2000–2006 period, different toxic episodes were detected at scallop production sites at concentrations or toxicities above PSMB regulatory limits (Table 3).

**Table 3.** Harmful algal blooms with implications for *Argopecten purpuratus* production.

| Species | Toxin | Syndrome | Geographical Location | Date | Duration | Source |
|---|---|---|---|---|---|---|
| *Pseudo-nitzschia australis* | Domoic Acid | Amnesic shellfish poisoning (ASP) | Bahía Inglesa (27°7′ S; 70°52′ W) | Sept. 2000 | 5–6 weeks | [73] |
| *Pseudo-nitzschia* sp. | Domoic Acid | Amnesic shellfish poisoning (ASP) | Bahía Inglesa (27°7′ S; 70°52′ W) | Oct. 2000 | 5–6 weeks | [73] |
| *Dinophysis acuminata* | Pectenotoxin | Diarrheic shellfish poisoning (DSP) | N.D. | 2005–2006 | Several episodes | [71,72] |
| *Alexandrium* sp. | Saxitoxin | Paralytic shellfish poisoning (PSP) | Bahía Mejillones (23°30′ S; 70°27′ W) | May 2006 | 2–3 weeks | [74] |
| *Alexandrium* sp. | Saxitoxin | Paralytic shellfish poisoning (PSP) | Bahía Tongoy (30°15′ S; 71°20′ W) and Bahía Guanaqueros (30°11′ S; 71°25′ W) | June 2006 | 1–2 weeks | [74] |
| *Pseudo-nitzschia australis* | Domoic Acid | Amnesic shellfish poisoning (ASP) | Bahía Inglesa (27°7′ S; 70°52′ W) | Nov. 2006 | 1–2 weeks | [71] |
| *Protoceratium reticulatum* | Yessotoxin | Diarrheic shellfish poisoning (DSP) | Bahía Mejillones (23°30′ S; 70°27′ W) | March 2007 | N.D. | [75] |

In 2000, two toxic episodes of amnesic shellfish poisoning (ASP) were recorded in Bahía Inglesa and led to a ban on harvesting, affecting landings and marketing. In 2006, a new episode of ASP was recorded over a short period, and moderate economic losses were recorded [71]. *Argopecten purpuratus* is a rapid domoic acid depurator (less than three days) to depurate 50% of the toxin; therefore, the time at which scallops are unsafe for consumers is usually very short (1–2 weeks), and the economic losses caused by ASP outbreaks in the aquaculture industry are moderate [71,73]. Paralytic shellfish poisoning (PSP) toxins have been detected in different bays dedicated to *A. purpuratus* aquaculture; however, to date, toxicity has never surpassed the regulatory limit, and therefore no harvesting ban has been implemented. Several episodes of diarrheic shellfish poisoning (DSP) were detected between 2005 and 2006. These blooms, together with the ASP episodes that occurred in 2006, forced authorities to issue several scallop harvest bans that prevented fish farmers from offering their production [71]. This situation restricted potential landings in that year and affected their marketing; however, the negative effect was offset by the maximum price recorded (i.e., USD 15.41 kg$^{-1}$) for *A. purpuratus* exports.

3.2.3. Environmental Disturbances Caused by Tsunami and Storms

Abrupt environmental disturbances (e.g., tsunami and storms) have affected Chilean coasts, with different results in scallop production. The 2010 tsunami with an epicentre located on the Maule coast, affected the area between the Araucanía and Valparaiso regions (i.e., central-southern Chile), devastated coastal areas and the livelihoods of artisanal fishers [76,77]. However, its effects did not reach the scallop production area. The earthquake and subsequent 2011 tsunami in Japan were considered the primary causes of the loss of production capacity due to the destruction of farming systems, boats, and facilities [1,5], a situation evidenced by the abrupt fall in production in successive years (Figure 7).

In August 2015, a storm affected the Chilean northern culture infrastructure [1,78], which generated the sinking of almost 90% of the boats in the bays of Tongoy and Guanaqueros. In September 2015, a tsunami was recorded off the coast of Coquimbo, very close to these same bays (i.e., where the highest production of *A. purpuratus* in Chile was officially recorded). The entanglement and dragging of suspended culture systems to the seabed caused production losses, and the time required for the normalisation of the culture systems reached three months (Lira, unpublished data). The environmental disturbances of 2011 and 2015 were pointed out as the last blow to an already struggling industry, forcing several companies to close (Figure 7) [1,5].

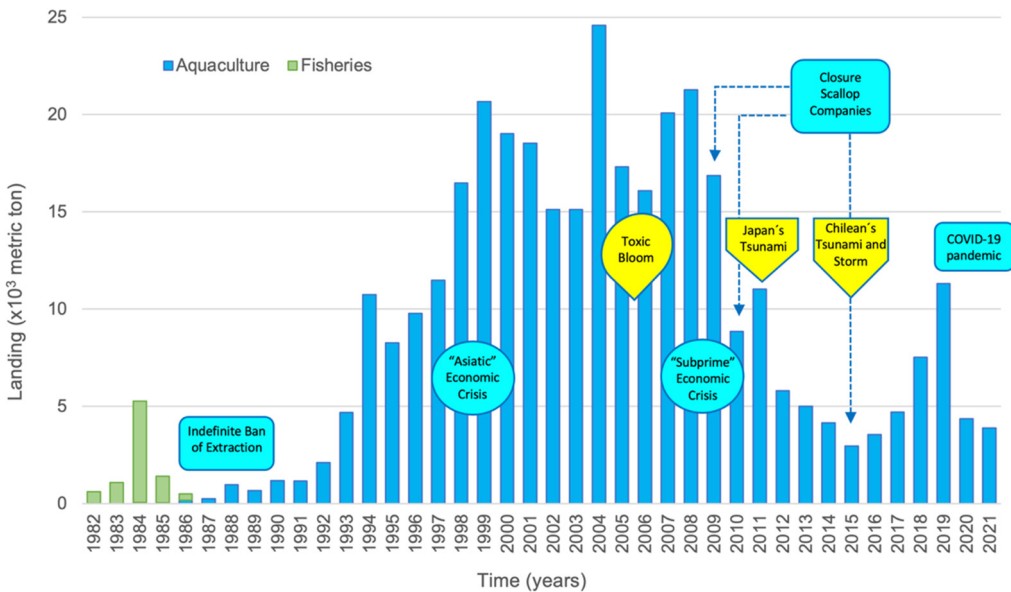

**Figure 7.** Influence of economic and environmental events on *Argopecten purpuratus* production.

### 3.3. Relationship between Economic and Commercial Factors and Scallop Production

#### 3.3.1. Market Structure

The evolution of Chilean exports, differentiated by market destinations, shows that the highest scallop export revenues occurred in 2006 and 2008 at USD 29.06 and 30.48 million, respectively [47]. However, the origin of these high revenues is different, since in the first case, it is due to a high export price of USD 15.41 kg$^{-1}$, while the second case is because of the high quantity exported (2.7 million tons). Scallop export revenues in 2004 were not among the highest due to a low export price of USD 10.15 kg$^{-1}$.

Since the beginning of the period analysed (1998 to 2006), France accounted for 90% of scallop exports. However, by 2012, exports to France had gradually declined until representing around 65% of total exports. In 2013, there was a shift in the demand of scallop, changing from French markets to the increasing demand from Spain. As a result, Spain concentrated more than 75% of Chilean exports in 2016. Exports to other markets remained at a low percentage, except in 2008, when Belgium reached 12.9% of the total exports (Figure 8). The reason for this change is that the markets of southern European countries are highly specialised in the consumption of bivalves with rooted cultural preferences of their residents (i.e., fresh consumption and in all its forms), differing from the residents of northern countries, who prefer the consumption of fresh fish and other seafood [79]. In particular, the French market demands scallops throughout the year despite showing seasonality in its national production and its consumption increases in price and quantity during key months of the year (e.g., summer and Christmas) [80]. In Spain, given the scarcity and high demand of an appreciated known local species called "*Zamburiña*" (*Mimachlamys varia*), *A. purpuratus* is marketed both in domestic markets and HORECA channels (hotels and restaurants) [81]. The presence of lipophilic toxins affected the production of scallops in Galicia, the main producer region in Spain, generating new episodes popularly known as red tide, which have caused the ban on the extraction of these bivalve molluscs. These environmental changes are the main reason that facilitated the entrance of the Chilean scallop in the Spanish market. The volume of scallop imports into the EU comes mainly from Asian and South Pacific countries and supplies the internal deficit of molluscs [79] generated by fisheries management systems (e.g., *Pecten maximus*) [80].

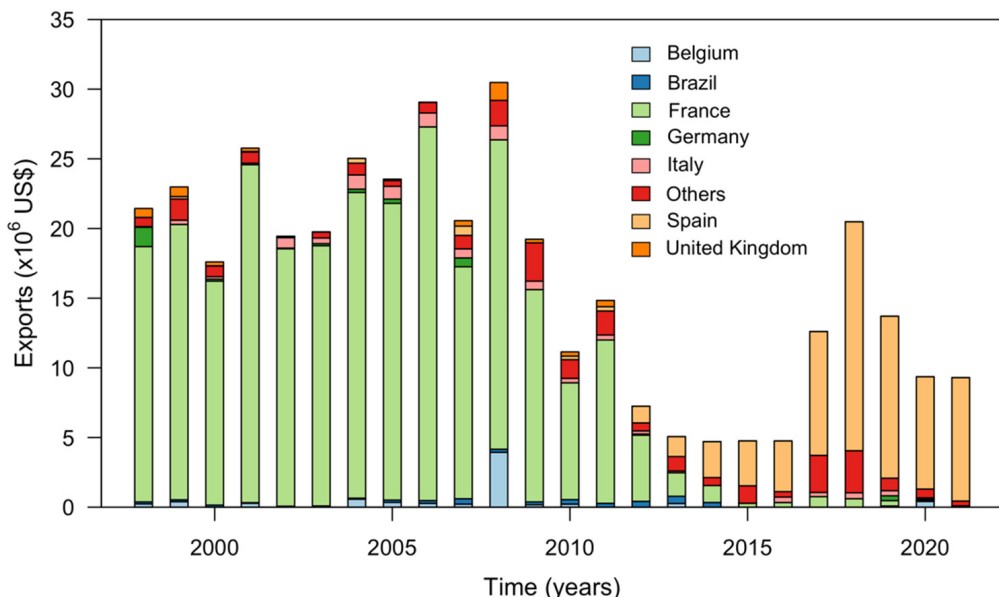

**Figure 8.** Distribution of Chilean exports by country of destination.

In addition, the historical trade trend of maintaining a single market as a buyer for a high percentage of Chilean scallop exports presented itself as an imperfect market, exposing the sector to a high dependence of a single buyer. Few buyers for the same product or species offered by several producers can be considered monopsonistic behaviour or oligopsony, which affects decision making due to the lack of alternative markets [82]. Another effect of an oligopsonistic market structure is the negative impact on selling prices due to the divergence between selling prices and competition prices [83,84], transferring part of the economic rent or surplus from the producer to the buyer [85]. The French market also has a negative effect on scallop prices, mainly when large volumes are exchanged with individual sellers [80]. Another market effect caused by disadvantages in product payment is the more significant negative impact generated on fishing cooperatives or small first-hand fish suppliers [86,87] because of their relatively high production costs. This situation of imperfect competition does not affect other food products in the same way. For example, agrifood products do not suffer significant effects in an oligopsonistic market configuration because producers can reallocate agricultural land between various commercial and non-commercial agricultural products [85], which gives them a higher elasticity concerning aquaculture.

### 3.3.2. Export Sales Prices

Figure 9 shows the evolution of the average export sales prices of scallops produced in Chile [47,49] and Peru [50] for the 1992–2021 period. The Chilean trend showed high variations from 1993 to 2005, ranging between USD 9.60 kg$^{-1}$ (2000) and USD 14.20 kg$^{-1}$ (1995). However, from 2006 onwards, a period of large fluctuations began that escaped this price band, reaching maximum prices of USD 15.41 kg$^{-1}$, and below this band, they fell to USD 8.21 kg$^{-1}$ and USD 9.00 kg$^{-1}$ in 2009 and 2015, respectively. The low prices recorded in these years have been pointed out as drivers of company closures in Chile (Figure 7).

For sales prices, the effects of both inflation and the dollar (USD) on the Chilean peso (CLP) exchange rate must be considered. For the inflation rate, for each CLP in 1997, it must be adjusted by 113.7% by 2021 https://calculadoraipc.ine.cl (accessed on 20 November 2022), that is, the CLP registers a loss of value over time by the same magnitude. In the exchange rate, the USD has strengthened against the CLP, registering equivalences of 419.31 and 759.07 CLP per USD in 1997 and 2021, respectively. Both adjustments produce offsetting results; on the one hand, the CLP loses value over time due to inflation, but this is offset by increased revenues due to USD sales to foreign countries. For domestic sales,

prices were affected only by inflation, resulting in a loss of competitiveness for companies targeting the domestic market.

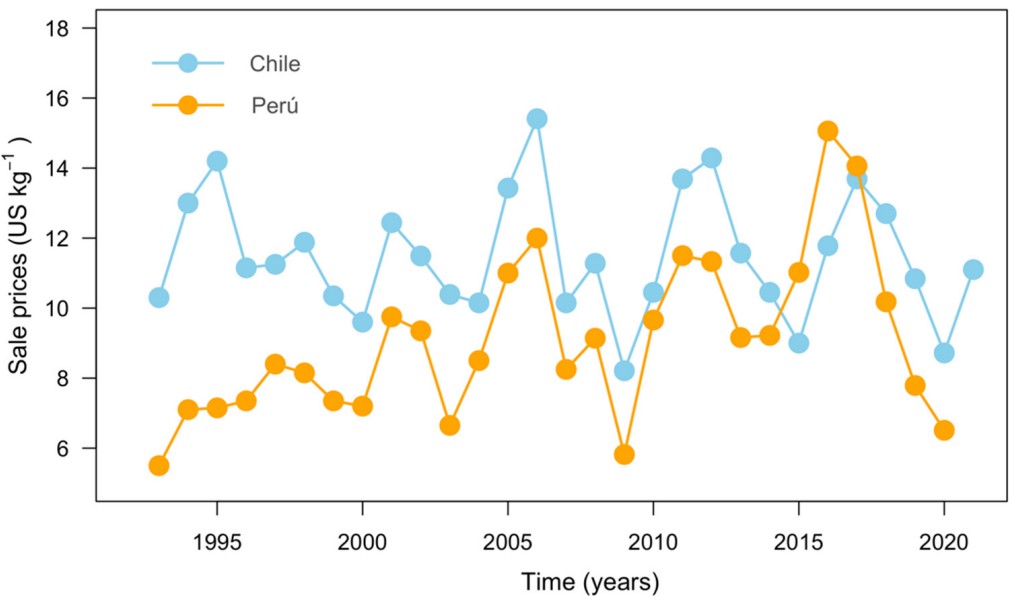

**Figure 9.** Average international prices for Chilean and Peruvian scallop production.

Comparing the international prices achieved by Chile and Peru, there is a marked difference until 1998, without significance (t = 6.575), which is based on the early presence of Chilean scallops in the French market and product quality [32,49]. The higher average price of Chilean scallops was maintained until 2014 (non-significance continues, t = 2.494), when Chilean producers decided to commercialise their products on the Spanish market. The transition and trade adjustments to the new market lasted until 2017, when the price leadership of *A. purpuratus* in international trade recovered. However, the average price reached in 2020 is noteworthy when scallops had to be sold by one of Chile's major producers because of the effects of the COVID-19 pandemic and the need for monetised production (Figure 9). Faced with this situation, some national producers turned to boosting domestic consumption in a fresh format, reaching approximately USD 0.5 per unit in tourist locations and exploring markets other than Spain.

The results from the augmented Dickey–Fuller (ADF) test showed a *p*-value < α (0.01 < 0.05), for which the null hypothesis is rejected, and it is concluded that the price of scallop in Chile does not present a unit root (stationary). Nevertheless, in Peru, the ADF test showed a *p*-value > α (0.167 > 0.05), for which the null hypothesis is not rejected, evidencing a unit root (non-stationary) (Table 4).

**Table 4.** Results of the augmented Dickey–Fuller (ADF) test from the time series of scallop prices in Chile and Peru between 1993 and 2020.

| Country | Dickey–Fuller Statistic Value | *p*-Value |
| --- | --- | --- |
| Chile | −4.38764 | 0.010 |
| Peru | −3.05993 | 0.167 |

The results from the Johansen cointegration test considering a significance level of 5%, when r = 0, the test statistic is greater than the critical value (22.81 > 19.96), which implies rejecting H0; therefore, there is statistical evidence of the presence of cointegration between the variables (Table 5). However, when testing H0 when r = 1, it is the critical value (CV) that exceeds the statistic (CV 9.24 > 7.50), so H0 is not rejected. In conclusion, by the trace test, the two variables analysed (Chile and Peru) present cointegration in a

maximum presence of 1 relationship, that is, the scallops from both countries are perfect substitutes on the global market.

**Table 5.** Results of the Johansen cointegration test from the time series of scallop prices in Chile and Peru between 1993 and 2020.

| Rank | Eigenvalue | Trace Test | 10% Critical Value | 5% Critical Value |
|---|---|---|---|---|
| 0 | 0.44522 | 22.81 | 17.85 | 19.96 |
| 1 | 0.25048 | 7.50 | 7.52 | 9.24 |

### 3.3.3. Effects of Peruvian Scallop Production

Figure 10 shows the Peruvian scallop production officially reported by the FAO. In Peru, the production showed an upward trend until 2010, when scallop aquaculture production stabilised. The production declined in Peru in 2012 due to unfavourable environmental conditions, such as the absence of an El Niño event, low oxygen levels in the water, and the high decomposition of organic matter [42], while in 2016, the production decrease was related to a strong El Niño event characterised by increased temperatures, lower salinities, and hypoxic conditions on the seabed [88]. Peru's highest production levels were 58,101 metric tons and 67,694 metric tons in 2011 and 2013, respectively. In the last decade, Peruvian production has exceeded Chilean production by 3 to 13 times. Currently, the officially registered aquaculture of *A. purpuratus* in Peru is ca. 50,000 metric tons.

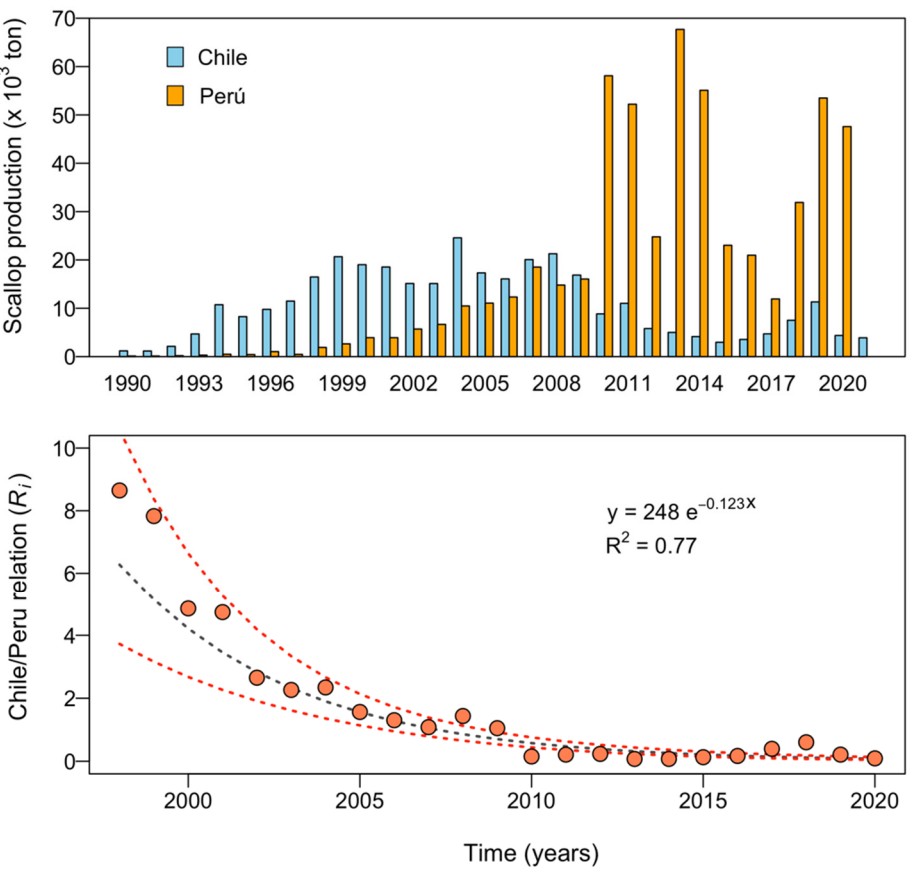

**Figure 10.** Chilean and Peruvian scallop production and their relationships.

Public and private efforts provided a productive and commercial boost to *A. purpuratus* aquaculture in Peru [42]. The incursion of Peruvian scallops into international markets is related to the stagnation of the sustained growth of scallop production in Chile until 2001. Between 2002 and 2009, scallop production remained in a stable, productive range



with low growth in both countries, except for Chilean landings in 2004, which reached their historical maximum of 24,647 metric tons. From 2009 to 2013, there was an indirect relationship between Chilean and Peruvian production; the first showed a gradual decrease in production, while the second showed an evident expansion (Figure 10). From 2014 onwards, both countries have experienced decreases and increases in scallop landings. Figure 10 compares and contrasts the scallop aquaculture production in Chile and Peru ($R_i$). Our results show a close inverse exponential relationship, i.e., Chile's production decreases while Peru's production increases. The regression analysis indicated a correlation factor ($R^2$) of 0.770.

### 3.3.4. Economic and Environmental Events

The relevant episodes in the international economy correspond to world economic crises, such as the Asian Crisis of 1997, which affected countries such as Indonesia, Malaysia, the Philippines, Thailand, and South Korea [89]; and the Subprime Crisis of 2007, which caused the most severe collapse of the banking systems ever recorded since the Great Depression of the 1930s [90] and a drop of 7% in seafood exports worldwide [91]. In principle, none of these crises seemed to generate effects on Chilean or Peruvian scallop production (Figure 7).

More recently, the COVID-19 pandemic has generated negative effects on the seafood sector and its supply and distribution chain [92,93], such as a reduction in the economic value [94] and demand [95] of seafood, risk to the food systems [96], and impacts on local livelihoods [97]. The aquaculture of *A. purpuratus* in Chile also experienced these impacts—in line with the finding by Amos et al. [98]—and, in 2020, national producers were forced to sell their production at a lower price (Figure 9) in order not to remain with unsold products in stock and generate sufficient income to sustain their production costs. This situation also led to changes in the ownership of national firms in 2022.

In the national context, together with the enactment of the LGPA in 1989, a national strategy was developed to promote scallop production in both production companies and small-scale fisheries. Both actions allowed the materialisation of a productive base of farming centres in the mid-1990s [14], which increased aquaculture landings in the country. As a result, the production capacity of the *A. purpuratus* industry in Chile at the end of the 2000s reached 500 million scallops. The closure of scallop-producing companies directly affected Chilean production, as 6 of the 11 companies formed by artisanal fishermen [99] and two of the largest companies, which together accounted for one-third of the scallop supply in Chile, were closed in 2009 [5,99]. In 2015, another large company closed, which was reflected in the decrease in production in the same year (Figure 7). The effect of small firms closing before large firms has been also reported by Young [87], who pointed out that artisanal fishers' cooperatives competing with large firms have low production efficiency and require government interventions. Such interventions were observed in the 1990s [31,100] and the years following environmental events (e.g., tsunami and storms); however, they aimed at replacing equipment and materials, not at improving production outputs, and thus, they did not generate significant increases in landings.

### 3.4. Challenges for Argopecten Purpuratus Industry in Chile

Currently, scientific support allows producers to culture *A. purpuratus* at all stages of production (Figure 2). However, to meet the scientific challenges in *A. purpuratus* production, it remains to be determined whether small-, medium-, and large-scale scallop aquaculture in Chile benefits from the comparative advantages created. The industry also faces sustainability challenges, which involve the producers as well as the environment. In relation to the producers, research must be promoted to provide solutions for continuous seed supply (which will help reduce idle production capacity) and quantity (to overcome environmental variations in recruitment) [13,22], provide more economical hatchery culture technologies [101], avoid the loss of genetic diversity and inbreeding depression [102], and improve seed resilience to the effects of climate change [103]. In addition, according to

Uribe et al. [2], the consolidation of a broodstock bank, the mechanisation and automation of daily tasks (e.g., handling, net maintenance, and post-harvest), and the implementation of information and navigation technologies (e.g., geographic information systems and drones) that reduce production costs were challenges raised by producers themselves. Environmental challenges include using renewable energy sources in farming systems [101,104] and the search for alternative uses of waste from the *A. purpuratus* industry to help reduce waste generation and promote the circularity of production processes [105].

After three decades of *A. purpuratus* industry in Chile, a greater role for producers in the execution and financing of R&D projects is required. The projects developed to date do not necessarily respond to the demands of the industry, but are due to other reasons, such as the interests of the researchers themselves, equipment and infrastructure, and the technical capacities already in place [36]. They also challenge science in terms of the political economy of production and management [106], governance [107], and the provision of information required to improve the competitiveness of scallop farming [12]. Furthermore, in line with Kumar et al. [108], to guarantee technology transfer, attention should be paid to the drivers that enable the final adoption of technological solutions, such as transfer methods, technology, and species characteristics, as well as economic, social, and institutional factors.

On the other hand, in patents, there is no evidence of the transfer of new technologies developed in universities or R&D centres to the private sector [109], a situation similar to the one observed in Chile [110]. Patents granted also appear as an imperfect indicator of the number of technological innovations made in developing countries [111], either because patent protection may stimulate foreign investment, but does not encourage technology transfer [112], or because firms sometimes opt for trade secrecy. Another challenge pointed out by Barton et al. [58], relevant to the *A. purpuratus* industry, is to determine whether R&D development in the Chilean industry is aimed at product innovation over sustainable regional development. The behaviour observed in both the execution of R&D projects and patenting can be described as the cause or effect of the productive model adopted by companies in Chile (e.g., mining and forestry) that base their competitiveness on comparative advantages and subsidies on built-in advantages [113,114].

To address the economic challenges, a distinction must be made between those related to the aquaculture industry and the ones resulting from the COVID-19 pandemic [95]. For *A. purpuratus* aquaculture in Chile, progress should focus on both the production strategies to reduce production costs [12], and commercial strategies to add value to scallops, penetrate the domestic market, which is currently covered by small-scale aquaculture [2], and evaluate new market niches arising from the decline of aquaculture of other species worldwide [115]. In addition, it is important to highlight innovations that have been attempted in different food systems that aim to change market structures, such as moving production chains forward by organising large-scale distribution to buy directly from producers, thereby reducing the bargaining power of exporters or large supermarket chains [116]. Another alternative is indicated by Mishra et al. in 2022 [117], who concluded that contract farming helps to reduce the price risk faced by farmers through contracts that offer insured marketing channels and with production risks shared between producers and contractors, also minimising the effects of a monopsonistic market configuration. Furthermore, the valorisation of waste from the *A. purpuratus* industry, through its use as a raw material in both the construction and food industries [105,118], generates competitiveness for producers due to the generation of additional income and environmentally reduces waste generation. Finally, given the rapid adaptation of small-scale fisheries sectors to abrupt shocks in the marketing system [94], it is necessary to assess small-scale production units that allow them to sell their products to other intermediaries or to the canning industry. For its part, the COVID-19 pandemic has resulted in emerging challenges, such as the scarcity of inputs, the impossibility to sell the product, the lack of transport for the supply of seafood [93], and others such as the low price of seafood, which without being the most important one, has a great impact on the subsistence of producers [93,98].

Transversally, there are challenges in having qualified human capital for aquaculture activities, strengthening the ecosystem of applied research and technological development [2], providing technical assistance [98], designing specific support instruments for small-scale aquaculture [2], and promoting the consolidation of *A. purpuratus* consumption in the country.

## 4. Conclusions

Scallop fishing has become a thriving aquaculture activity that has allowed an intensive harvesting and commercialisation of the species; aquaculture yields up to five times the maximum obtained in its fishing stage. A series of endogenous and exogenous factors promoted a boom in the production of *A. purpuratus*; however, given the productive conformation of the industry, they did not allow its sustainability. The dynamics of the scallop production in Chile is illustrated as a plateau-like collapse [119]. Currently, scallop production in Chile is sustained by a third of the companies that were established in its heyday, which points to the challenge of moving from an industry focused on prices and production volumes to one that develops technology to reduce production costs, add value to its products, and develop strategies for sustainable production, as well as to strengthen the domestic market and exports.

Our findings broaden the understanding of how economic, technological, and environmental effects greatly influenced the decline of scallop production in Chile, which can be complemented by studies that allow the establishment of quantitative models that consider both the continuous and discrete variables addressed in this study. They also serve as a basis for public and private decision makers to address the current and future challenges facing the *A. purpuratus* industry, as well as those triggered by the effects of the COVID-19 pandemic.

**Author Contributions:** Conceptualization, J.B. and G.Á.; methodology, J.B., G.Á., P.A.D. and T.G.B.; software, J.B. and P.A.D.; validation, J.B., G.Á., P.A.D., E.U., R.S. and S.V.; formal analysis, J.B., G.Á. and P.A.D.; investigation, J.B., G.Á., E.U., R.S., T.G.B., G.L., H.P., A.H., R.G.-Á. and J.C.-V.; resources, J.B., G.Á., S.V., T.G.B., G.L., H.P., A.H., R.G.-Á. and J.C.-V.; data curation, J.B., G.Á. and T.G.B.; writing—original draft preparation, J.B., G.Á., P.A.D., S.V., T.G.B., G.L., H.P. and A.H.; writing—review and editing, J.B., G.Á., P.A.D. and S.V.; visualization, J.B. and P.A.D.; supervision, J.B. and G.Á.; project administration, J.B., E.U., and R.S.; funding acquisition, J.B., G.Á. and H.P. All authors have read and agreed to the published version of the manuscript.

**Funding:** This study was funded by the Fondo de Investigación Pesquera y Acuicultura (FIPA) through the project "Análisis del desarrollo histórico y colapso del cultivo del ostión del norte como herramienta para el re-impulso de la actividad en la III y IV Regiones, FIPA 2017-12" and "ANID + FONDEF/PRIMER CONCURSO INVESTIGACIÓN TECNOLÓGICA TEMATICO EN SISTEMAS PESQUERO ACUICOLAS FRENTE A FLORECIMIENTOS ALGALES NOCIVOS FANS IDeA DEL FONDO DE FOMENTO AL DESARROLLO CIENTÍFICO Y TECNOLÓGICO, FONDEF/ANID 2017, IT17F10002" developed within the framework of a cooperation agreement between the Consellería do Mar, Xunta de Galicia, Spain, and the Universidad Católica del Norte, Chile. Patricio A. Díaz is funded by the Centro de Biotecnología y Bioingeniería (CeBiB) (PIA project FB0001, ANID, Chile). The APC was funded by Centro de Innovación Acuícola AquaPacífico and Universidad Católica del Norte.

**Data Availability Statement:** Not applicable.

**Acknowledgments:** We thank Manuel Andrade from the Subsecretaría de Pesca y Acuicultura (SUBPESCA) and the staff of the ProChile–Coquimbo Regional Office for their advice and access to data. We also thank Juan Enrique Illanes, Mario Fajardo, Rodrigo Poblete, and Osvaldo Miranda from the Universidad Católica del Norte (UCN) for additional information, research guidance, and suggestions, which improved the manuscript.

**Conflicts of Interest:** The authors declare no conflict of interest.

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
