# Peer review of "Disentangling Environmental, Economic, and Technological Factors Driving Scallop (Argopecten purpuratus) Aquaculture in Chile"

_fishes, doi:10.3390/fishes7060380_

Round 1

Reviewer 1 Report

Generally, I found this an interesting paper covering a lot of potential issues relating to production of scallop aquaculture in Chile. As an overview of these issues, the paper is good, although some of the arguments are not convincing based on the evidence presented. This is particularly the case with production and climate (La Nina/El Nino), where the apparent relationship described in the text does not seem to match with the figures (where little of no relationship seems apparent).

Economic drivers are also not apparent. The prices seem to be in nominal rather than real terms (i.e., adjusted for inflation). If converted to real terms, I would expect prices in Figure 9 to decline over time, which may provide greater evidence of the impact of price on production. Given the importance given to the two price series (Chile and Peru), tests for stationarity and a simple co-integration analysis would be useful to determine the strength of the relationship between them, and determine if they are perfect or imperfect substitutes on the global market.

With so many potential pressures on the production, there may be benefits in undertaking a more quantitative analysis to actually disentangle these effects (as the qualitative description keeps them very much entangled). For example, regressing production against the climate variables, the environmental shocks outlined in Figure 7, and the price (expressed in real terms) and ideally some index of production costs (if available – costs are mentioned on page 17). Dummy variables to capture the impacts of Covid could also be included. Potentially, R&D expenditure and the number of patents could also be added to the model. A time trend would also be a useful addition to capture any disembodied technical change.

Without such an analysis, it is very hard to see the effect that all of these factors had (or did not have) on production.

Reviewer 2 Report

This paper examines the environmental, economic and technical factors affecting the development of the scallop fishery in Chile during the last 42 years. You base your analysis exclusively on secondary data drawn from an “exhaustive literature review” of 309 papers in scientific literature; 159 patents, and various state reports and historical accounts since 1980 (lines 109-111). You conclude that the main challenge is to reduce costs of production, not least by embracing renewable energy and by developing a circular resource use and waste disposal loop.  

 In my view, your paper is publishable virtually as it stands. It is well written in good English, and coherently presents its material and arguments. It focuses on an important issue - how to improve scallop production in Chile – and it provides a comprehensive analysis of the scientific/technical factors; the ecological and environmental factors; the economic factors; and the Covid factors which have affected scallop production in Chile. However, I would like you to respond to the following observations before I recommend publication.

(1) In the methodology section, how many state reports and historical accounts were used?

 (2) The patents section 3.1.3 is odd for two reasons. First, it hardly mentions Chile, concentrating on the number of patents held by countries across the world. Second, there is no explanation of why patents are relevant to the argument in the paper: there is no answer to the question ‘how have patents affected the production of scallops in Chile?’

 (3) The Conclusion is not very specific in its recommendations:

 “the challenge of moving from an industry focused on prices and production volumes to one that develops technology to reduce production costs, add value to its products and develop strategies for sustainable production and to strengthening the domestic market as well as exports” (lines 573-576)

 These recommendations are generic, and could be applied to almost any kind of industry. Precisely how do you expect the scallop industry in Chile to use technology to reduce production costs; to add value to its products; to develop strategies for sustainable production; and to strengthen its domestic and export markets? 

Round 2

Reviewer 1 Report

I would still liked to have seen more quantitative modelling but can accept this can be done as a follow up paper. The addition of the cointegration analysis was good, and a useful addition. A plot of the real prices (i.e. correcting for inflation) would also still be useful, but I am satisfied that the overall effects of these changes have been mentioned in the manuscript.